# Whole-Genome Shotgun Sequencing from Chicken Clinical Tracheal Samples for Bacterial and Novel Bacteriophage Identification

**DOI:** 10.3390/vetsci12020162

**Published:** 2025-02-12

**Authors:** Klaudia Chrzastek, Bruce S. Seal, Arun Kulkarni, Darrell R. Kapczynski

**Affiliations:** 1Exotic and Emerging Avian Diseases Research Unit, U.S. National Poultry Research Center, Agricultural Research Service, U.S. Department of Agriculture (USDA), 934 College Station Road, Athens, GA 30605, USA; 2Center for Translational Antiviral Research, Institute for Biomedical Sciences, Georgia State University, Atlanta, GA 30303, USA; 3Biology Program, Oregon State University—Cascades, 1500 SW Chandler Avenue, Bend, OR 97702, USA; bruce.seal@osucascades.edu; 4Georgia Poultry Laboratory Network, 3235 Abit Massey Way, Gainesville, GA 30507, USA; akulkarni@gapoultrylab.org

**Keywords:** shotgun, metagenomics, *Ornitobacterium rhinotracheale*, prophage, genome annotation, DNA sequencing, bacteria sequencing, ORT

## Abstract

The diagnostic testing of clinical samples is critical when providing client care to food animals. Current submission procedures rely on the culturing of samples to determine the causative pathogen contained within them. Newer molecular technologies, such as whole-genome sequencing (WGS), offer a faster and more precise mechanism to determine pathogen identification within a sample. In this study, we directly identified a bacterial pathogen and prophage virus from the direct DNA sequencing of clinical samples of chickens undergoing respiratory distress. The results were obtained days after submission rather than the weeks that it can take to culture organisms in broth or on agar. In addition, a complete “tree of life”, based on DNA sequencing, can be obtained with these methods. Taken together, WGS can be a favorable method for pathogen identification in clinical cases.

## 1. Introduction

Shotgun metagenomics is one of the high-throughput sequencing (HTS) methods that allows for untargeted genomic analysis, ideally of all the microbes in a sequenced sample. Shotgun whole-genome sequencing (sWGS) can be used to identify the taxonomic composition of a microbial community in the sample, as well as allowing for the functional annotation of the microbial genes and, in some cases, to recover a whole genome sequence [1].

Bacterial WGS is not only useful for bacterial identification, but it is also a promising tool for enhancing clinical bacteriology, for surveillance, and/or for the detection of drug-resistant mutations such as investigating the methicillin resistance of *Staphylococcus aureus* [2,3,4]. WGS can also be applied for the analysis of foodborne outbreaks, such as listeriosis and salmonellosis, or as a powerful tool for bacteria surveillance [5,6,7,8,9,10]. Furthermore, by applying WGS, it is possible to track trends associated with pathogen virulence and antimicrobial resistance, as well as commonly used single-nucleotide polymorphism (SNP) analysis, core genome multi-locus sequence typing (cgMLST), or whole-genome MLST [11,12,13,14,15,16]. Recently, bacterial WGS was also successfully implemented to the poultry industry, particularly to characterize *Salmonella* spp. and *Campylobacter* spp. [17,18,19,20] or *Escherichia coli* [21].

*Ornithobacterium rhinotracheale* (ORT) is Gram-negative bacterium of worldwide distribution in the commercial poultry industry. It is considered as a possible causative agent of complex respiratory diseases in birds, associated with respiratory illness, reduced egg production, and mortality [22]. ORT can infect a wide range of birds, including domestic species such as chicken, turkey, duck, ostrich, or quail [23]. The first complete genome sequence of *Ornithobacterium rhinotracheale* (strain ORT-UMN 88) was isolated from a pneumonic turkey lung in the USA [24]. The complete first genome sequence of ORT isolated from chickens was submitted in 2023 (GenBank database: NZ_CP102745.1). Although there is an increased number of bacterial and other organisms’ genomes sequenced and submitted to GenBank, bacteriophages (phages) are not keeping pace with the current trend, which is most probably due to challenges in prophage identification or proper phage annotation. Most of the phages or prophages remain novel and not classified; therefore, their BLAST-based analyses continually face challenges in identification [25,26,27]. Bacteriophages have important ecological and evolutionary impacts on their bacterial hosts as they can insert their own or foreign DNA into bacterial cells. Bacteriophages are also able to transduce genes to bacterial genomes, including antibiotic resistance genes. Prophages are also able to multiply and kill their target host bacteria [28,29].

In this study, a whole-genome shotgun sequencing approach was applied to chicken tracheal swab samples to potentially identify *Ornithobacterium rhinotracheale* (ORT) sequences among poultry with respiratory disease (DDBJ/ENA/GenBank under the accession VWRM00000000). Multi-locus sequence typing (MLST) along with detection of antibiotic-resistant genes was also performed. Subsequently, a novel bacteriophage, presumably a prophage, encoding typical bacteriophage genes and two endolysins in the sequence was also identified in the same clinical sample.

## 2. Materials and Methods

### 2.1. Clinical Samples

Tracheal swabs (N = 6) in BHI broth (MilliporeSigma, Burlington, MA, USA) were obtained from 30-week-old mixed breeders exhibiting mild clinical signs of respiratory distress in North Georgia and were stored at −20 °C until processed for genomic DNA extraction. The samples were collected during routine veterinary health checks at the commercial farm.

### 2.2. DNA Isolation and Sequencing

Genomic DNA (gDNA) was extracted directly from clinical trachea swab samples using a commercially available DNA isolation kit (DNeasy blood and tissue kit, Qiagen, Germantown, MD, USA), according to the manufacturer’s instructions. Genomic DNA concentration was measured using the Qubit^®^ dsDNA HS Assay Kit (Life Technologies, Carlsbad, CA, USA). In total, 1 ng of DNA was used to prepare a library using the Nextera XT Fragment Library Kit (Illumina, San Diego, CA, USA). Libraries were analyzed with a high-sensitivity DNA chip on the bioanalyzer (Agilent Technologies, Corvallis, OR, USA) before being loaded onto the flow cell of the 500-cycle MiSeq Reagent Kit v2 (Illumina, San Diego, CA, USA) and undergoing pair-end sequencing (2 × 250 bp). Whole-genome sequencing was performed using the Illumina MiSeq platform.

### 2.3. Analysis of Sequencing Results

A workflow applied to perform an analysis of the sequencing reads is depicted in Figure 1. The multistep workflow presented here allows for (i) pathogen detection by de novo assembly based species identification and metagenomics; (ii) MLST and antibiotic resistant analysis using BLAST-based approaches; and (iii) reference-guided assembly to generate a consensus sequence (Figure 1).

All bioinformatic tools used in this study are freely available. Briefly, the quality of sequencing reads was assessed using FastQC ver. 0.11.5; the reads were then quality trimmed with a Phred quality score of 30 or more, in addition to low-quality end trimming and adapter removal using Trim Galore ver. 0.5.0 (powered by Cutadapt) [30]. De novo assembly was performed using the Velvet de novo Assembler installed in the Illumina BaseSpace app (Version 1.0.0) (Illumina BaseSpace app default settings), the SPAdes de novo assembler (version 3.10.1) (k-mer 33, 55, and 77), and the IDBA-UD de novo assembler (version 1.1.3) (min. k-mer 35 and max. k-mer 95); the quality was assessed using QUAST (version 5.0.2) [31,32,33]. In parallel with de novo assembly, Kraken Metagenomics Classifier Version: 1.0.0 [34] was used to analyze sequence sets. De novo-assembled reads were then used for a k-mer-based species identification using KmerFinder 3.1 [35,36,37] and multi-locus sequence typing (MLST 2.0) [38] and were screened to determine antibiotic-resistant genes using the online Comprehensive Antibiotic Resistance Database (CARD 3.0.2) and Resistant Gene Identifier (RGI 5.0.0) [39]. Scaffold_builder [40] was used for reference-based scaffolding and *Ornithobacterium rhinotracheale* ORT-UMN 88 (GenBank CP006828) was used as the reference genome. The NCBI’s Prokaryotic Genomes Annotation Pipeline (PGAP) annotated the genome sequence. The nucleotide sequences of the 16S rRNA of well-characterized reference strains and isolates from GenBank along with the 16S rRNA obtained in this study were aligned using MUSCLE [41]. ML phylogeny was generated with MEGA7 based on the Hasegawa–Kishino–Yano nucleotide substitution model [42,43]. Bootstrap support values were generated using 1000 rapid bootstrap replicates.

Putative prophage sequences were identified using Prophage Hunter [44] and PHASTER [45,46]. The identification of genes was achieved by predicting open reading frames (ORFs) using ATG, GTG, and TTG start codons with a minimum nucleotide length of 50 bp using GeneMark for prokaryotes [47,48] and the most recent version of ORF Finder software [49]. Next, the ORFs were manually re-analyzed by performing a BLASTP search against the GenBank database.

## 3. Results

### 3.1. Sequencing Directly on Clinical Samples—Identification of Ornithobacterium rhinotracheale (ORT)

After shotgun WGS, we found ORT and a putative prophage candidate in one of the clinical samples. For de novo assembly, a total of 141,868 paired-end reads were assembled using Velvet, IDBA-UD, and SPAdes de novo assemblers into 4960, 3516, and 7895 contigs, respectively. The length of the longest contig and the total length of assembly was 33,840 bp (Velvet), 68,857 bp (IDBA-UD and SPAdes), 2,255,318 bp (Velvet), 3,084,290 bp (IDBA-UD), and 3,551,185 bp (SPAdes), respectively. A detailed statistic set of the de novo assembly is shown in Table 1.

For species identification, we used de novo-assembled contigs as an input for KmerFinder 3.1 and pre-processed sequencing reads for Kraken data analysis. Both platforms identified *Ornithobacterium rhinotracheale* (ORT) in the sample. Detailed statistics of species identification using KmerFinder 3.1 are shown in Table 2, whereas the distribution of the reads classified by Kraken for clinical samples is shown in Figure 2. Out of 141,868 paired reads, 38,895 reads were classified into the “Bacteria” domain by Kraken, and 70% of reads were classified as ORT (27,369 sequencing reads).

### 3.2. MLST Analysis of Ornithobacterium rhinotracheale (ORT) Genome

For further species characterization, we applied BLAST-based approaches (MLST and antibiotic resistance gene screening). MLST analysis demonstrated an identical MLST type of the clinical isolate as compared to ORT sequences deposited in the GenBank CP006828.1 (ORT-UMN 88; pure culture turkey isolate) (Table 3).

Antibiotic resistant gene analysis identified tetQ (97.2% identity of matching region, and 97.1% of reference sequence length), aminoglycoside-(3)-acetyltransferase IV gene (AAC(3)-IV), aminoglycoside antibiotic inactivation (99.5% identity of matching region and 89.2% of reference sequence length), and macrolide resistance, as well as the ermX gene (70.8% of reference sequence length) within the bacterial genome (Table 4).

However, no resistance genes were found for Rifampicin, Colistin, Fosfomycin, Sulphomide, Trimethoprim, Fluoroquinolone, Oxazolidinone, Beta-lactam, Nitroimidazole, or Glycopeptide.

### 3.3. Reference-Based Ornithobacterium rhinotracheale (ORT) Genome Assembly

Detailed statistics for BWA-MEM reference mapping to *Ornithobacterium rhinotracheale* ORT-UMN 88 (GenBank CP006828) are provided in Table 5.

Out of 141,868 paired-end sequencing reads, 64,195, 72,130 and 75,136 reads were mapped to the reference genome using Velvet, IDBA-UD, and SPAdes assemblers, respectively. Velvet assembled 1.8 Mb of contigs, whereas IDBA-UD and SPAdes assembled 2.0 Mb of contigs. Velvet assembled with an NG50 of 1605 bp compared to an NG50 of 3340 bp for IDBA-UD (Table 5). The genome fraction covered by sequencing reads was from 76% (Velvet) to 86% (IDBA-UD and SPAdes) (Figure 3A). The number of genes was 809 complete and 1086 partial for Velvet; 1296 complete and 743 partial for IDBA-UD; and 1264 complete and 767 partial for SPAdes (Figure 3B). The average depth of coverage was similar for all three assemblers used in this study, but Velvet produced less misassembles and mismatches compared to IDBA-UD and SPAdes (Figure 3C).

### 3.4. Phylogenetic Analysis of 16S rRNA of Ornithobacterium rhinotracheale (ORT)

The phylogenetic analysis of the 16S rRNA gene demonstrated 99.1–100% pairwise identity of clinical chicken ORT sequenced here (ORT3331-2017) to ORT sequences deposited in the GenBank. Specifically, the gene clustered with chicken and turkey ORT isolates serovar A and serovar B (Appendix A).

### 3.5. Identification of a Novel Prophage

Interestingly, after the de novo assembly of sequencing reads, the longest contig of 68 kb did not match with any ORT genome and was thus used for further investigation. Along with the 68 kb contig, all others that did not align with the ORT genome (N = 72,237,435 bp in total) were used as an input for Kraken metagenomics. Kraken assigned 21 of them to the “Virus” domain, 8 as *Shigella* phage ShFl1 type, and 7 contigs as a *Pseudomonas* phage type (Figure 4).

The largest contig of 68 kb was then used as an input for phage screening using the Prophage Hunter and PHAST tools. A 27 kb length active prophage candidate, containing 22 CDS, was predicted by Prophage Hunter (Table 6) as compared to PHAST, which predicted a 24.7 kb prophage region length containing 18 proteins (Table 7).

Phage-related proteins identified as incomplete prophages by PHAST are shown in Figure 5. The number of hypothetical proteins in the region without a match in the database was nine.

Interestingly, two regions that code for an endolysin were found in the analyzed prophage sequence. The closest related phages based on E-value and Bitscore were *Acinetobacter* phage Presley, *Salmonella* phage FSL SP-076, *Salmonella* phage FSL SP-058, *Escherichia* phage Pollock, and *Vibrio* phage JSF3, although the sequence identity was less than 72% (Table 8).

The prophage sequence obtained from a chicken tracheal swab sample was characterized as most closely related to an N4-like Prophage Chicken Bacterial Metagenome based on BLAST analysis. All ORFs predicted were next manually re-analyzed by a BLAST search. BLAST analyses of annotated ORFs from the putative prophage sequence resulted in a variety of potential bacterial-encoded and principally bacteriophage-encoding genes, indicating that the sequenced DNA was a potential prophage.

This whole-genome shotgun project has been deposited in DDBJ/ENA/GenBank under accession number VWRM00000000, *Ornithobacterium rhinotracheale* ORT3331-2017. The version described in this paper is version VWRM01000000. All read sequences were deposited in the Sequence Read Archive (SRA) of the NCBI (http://www.ncbi.nlm.nih.gov/sra) on 2 February 2022 under accession number SRR9903547. The prophage sequences were submitted under GenBank accession number MT025940.

The timeline for the detection and characterization of the clinical sample was approximately 3 days (including in silico analysis and Illumina Miseq run—49 h).

## 4. Discussion

The rapid diagnostic identification and characterization of infectious pathogens, as well as determining their antibiotic susceptibility, are essential to guide therapy, so that directed treatment can be initiated. Conventional clinical microbial diagnostic methods are mainly based on the culturing of samples on different agar plates, followed by antibiotic susceptibility testing. This procedure usually takes around 4 days (for culturing, species identification, and susceptibility testing) and even longer for further characterization on a case-by-case basis. Using WGS directly on isolates and clinical samples can reduce the time necessary to diagnosis and identify pathogens by one-third.

In this study, shotgun WGS was applied to chicken clinical samples and after bioinformatics analysis of sequencing reads, *Ornithobacterium rhinotracheale* (ORT) and novel prophages were identified in one of the samples without the need for bacteria culture. ORT is a bacterium causing respiratory illness in poultry, contributing to economic loss in the poultry industry worldwide [22,26,28,50,51,52,53,54,55]. In addition, co-infection with ORT and several other respiratory infectious agents such as *Chlamydia psittaci*, influenza virus subtype A, Newcastle disease virus (NDV), Infectious bronchitis virus (IBV), and avian metapneumovirus (aMPV) may occur and contribute to high mortality rates in poultry farms [56,57,58,59,60]. Different typing methods have been applied to characterize ORT strains, including serotyping; molecular typing, such as multi-locus enzyme electrophoresis (MLEE), repetitive sequence-based polymerase chain reaction (rep-PCR), and 16S ribosomal RNA (rRNA) gene analyses; random amplified polymorphic DNA (RAPD); pulsed-field gel electrophoresis (PFGE); or multi-locus sequence typing (MLST) [52,53,61,62,63,64,65]. The MLST analysis along with k-mer identification performed in this study allowed for the fast and easy identification of ORT in the sample. Genus and species identification and the characterization of ORT took approximately 3 days in this study. It is important to note that WGS also allowed antibiotic resistant gene analysis and the detection of the tetQ tetracycline gene. Due to difficulties in culturing and a lack of ORT genome sequences in the database, the application of WGS to clinical samples is a very promising tool for ORT characterization and may increase the number of positive detections and available genome datasets.

A major challenge of the WGS of clinical samples is the genome assembly of a detected pathogen. We compared the efficiency of the Velvet, IDBA-UD, and SPAdes genome assemblers to generate the ORT genome sequence. Although all assemblers allowed for species detection and characterization, IDBA-UD and SPAdes covered a higher genome fraction compared to the Velvet assembler. However, the number of mismatches produced was lower after Velvet assembly compared to IDBA-UD and SPAdes. As demonstrated here, it is important to apply multiple different de novo assemblers to the clinical sample generation of the final consensus sequence when the content is unknown. Our simple and freely available bioinformatic workflow, as presented here (Figure 1), can be used as guidance for any species characterization where the main focus is on the DNA sequencing of clinical samples.

Interestingly, in addition to ORT identification, we also discovered a novel prophage/bacteriophage in the same sample. Identified bacteriophage-encoded genes included an HNH endonuclease signature which is found in viral, prokaryotic, and eukaryotic genomes. Interestingly, these gene products play key roles during phage DNA packaging along with terminase and portal proteins [66], which are also identified in the putative prophage sequences. The HNH endonuclease gene was reported in the deep-sea thermophilic bacteriophage *Geobacillus* virus E2 (GVE2), wherein the protein reportedly possesses a typical ββα-metal fold and Zn-finger motif like those of HNH endonucleases encoded by other bacteriophages [67]. A gene encoding a putative portal protein was detected similar to the one in *Escherichia* phage Pollock (YP_009152169.1) reported as an N4-like podophage [68]; a putative terminase gene was also detected similar to the *Escherichia* phage vB_EcoP_PhAPEC5 (YP_009055580.1) from an avian pathogenic *Escherichia coli* (APEC) host causing colibacillosis in poultry [69]. Other putative prophage protein-encoding genes included those necessary for viral genome replication. Specifically, a DNA polymerase I gene encoding 3′-5′ exonuclease and polymerase domains, like one reported in *Salmonella* phage FSL SP-058 (YP_008239451.1) and related viruses [70], was detected in the putative prophage sequence. This putative Pol gene was also related to that encoded by the *Erwinia* phage vB EamP that infects plant pathogens [71]. Also detected in the putative prophage sequences was a reported RNA-dependent RNA polymerase P1-P2 fusion/replicase protein gene found in plant Luteoviruses (pfam08467) [72] and reported among *Erwinia* bacteriophages [71]. Putative genes encoding a thymidylate synthase similar to *Vibrio* phage SSP002 (YP_009598659) [73] and a DNA primase similar to the N4-like *Escherichia* phage Pollock (YP_009152152.1) [68] were also detected in the putative prophage sequences. Other bacteriophage protein-encoding genes detected in the putative prophage sequences included a potential phage tail collar domain found among *Pasteurella* genomes (SPY33204, WP_096742905). These proteins are structural components of the bacteriophage tail fiber baseplate common among the *Autographiviridae* family [74]. A gene encoding a potential bacteriophage major capsid protein similar to N4-like Pseudomonas viruses [75] was also detected that indicates the prophage sequence is indicative of a podovirus or the proposed “*Enquartavirinae*” subfamily [76]. A potential tape measure protein-encoding gene was detected similar to *Erwinia* phage vB EamP-S6 was also reported as a member of the Autographiviridae [71,74]. As with many bacteriophage genomes, hypothetical proteins with no known function were detected as encoded in the sequence and these had similarity to other phage genes such as *Escherichia* phage Pollock (YP_009152178) and *Podoviridae* sp. ctLUJ1 [77], all of which are Podoviruses with short tails. Although a search of *O. rhinothracheale* genomes revealed no reported bacteriophage sequences in the database, several phage-associated genes have been identified in the candidate *O. hominis* [78]. These bacteriophage genome islands range from >10 kb to 30 kb, representing genes associated with potential tailed bacteriophages. Consequently, the data reported herein present a similar genomic discovery for *O. rhinothracheale*.

Several genes were identified that encoded potential lysins, including a C-terminal portion of an N-acetylmuramoyl-L-alanine amidase similar to a gene reported in *Flavobacterium* sp. BBQ-12 (WP_128196763). An aminopeptidase N (WP_044831623) and a putative peptidoglycan DD-metalloendopeptidase family protein-encoding gene was also detected in the prophage sequence. The latter protein includes soluble lytic transglycosylases (SLTs) in the lysozyme superfamily involved in the hydrolysis of beta-1,4-linked polysaccharides that could potentially be utilized as potential antimicrobials [79]. Other potential lytic proteins included a putative metallopeptidase domain also reported in *Salmonella* phage FSL SP-076 (YP_008240169) and *Escherichia* phage Pollock (YP_009152135), as well as a potential C-terminal chaperone domain of a bacteriophage endosialidase (AIX45815), which is part of the capsule-degrading tailspikes [80]. Potentially, these gene products could be utilized as antimicrobials or diagnostic molecular tools [81].

## 5. Conclusions

In summary, direct whole-genome shotgun sequencing was applied to chicken clinical samples and *Ornithobacterium rhinotracheale*, along with a novel bacteriophage being detected in the sample. We presented a simple and freely available workflow for DNA sequencing. This study demonstrates that WGS is a powerful tool that can reduce diagnostic time, overcome the limitations of culture-dependent identification of infectious pathogens, and has potential in clinical microbiology by identifying antibiotic resistance genes. Shotgun sequencing also allows for novel pathogen discovery and aids in understanding the co-existence of different microorganisms in the same niche. There are, of course, many challenges that clinical WGS microbiology will need to overcome in the future, including host DNA contamination, the development of bioinformatic workflows, developing standard operating procedures and data storage, etc. However, as clinical diagnostics are growing rapidly, the development of new techniques to more rapidly identify etiologic agents is required.

## Figures and Tables

**Figure 1 vetsci-12-00162-f001:**
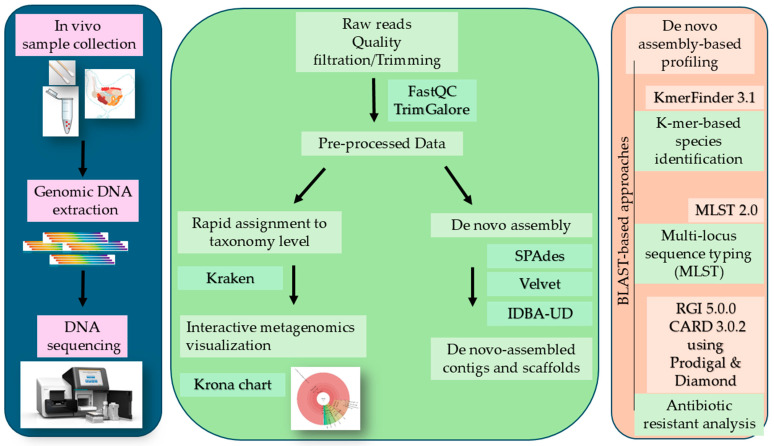
Schema of whole-genome sequencing workflow applied to clinical samples. The first step consists of sample collection, genomic DNA extraction, and DNA sequencing (blue chart). Metagenomics and de novo assembly are shown in green; these consist of the quality processing and trimming of sequencing reads followed by Kraken metagenomics read assignment to taxonomic levels and the de novo assembly of sequencing reads using SPAdes, Velvet, and the IDBA-UD assembler (green chart). For further characterization, we used BLAST-based approaches (orange chart).

**Figure 2 vetsci-12-00162-f002:**
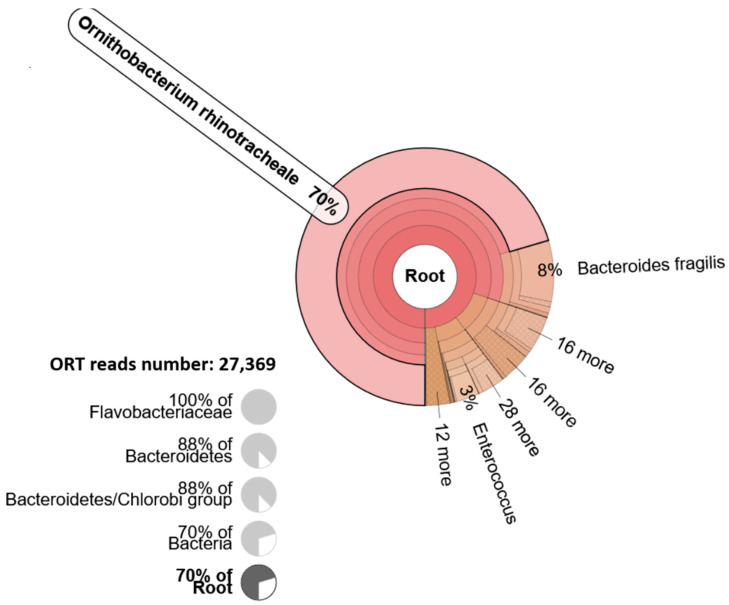
Taxonomic distribution of sequencing reads obtained from a representative chicken clinical sample classified by the Kraken metagenomics tool. The results are visualized using a Krona chart. The chart is placed at the root; the species *Ornithobacterium rhinotracheale* is selected.

**Figure 3 vetsci-12-00162-f003:**
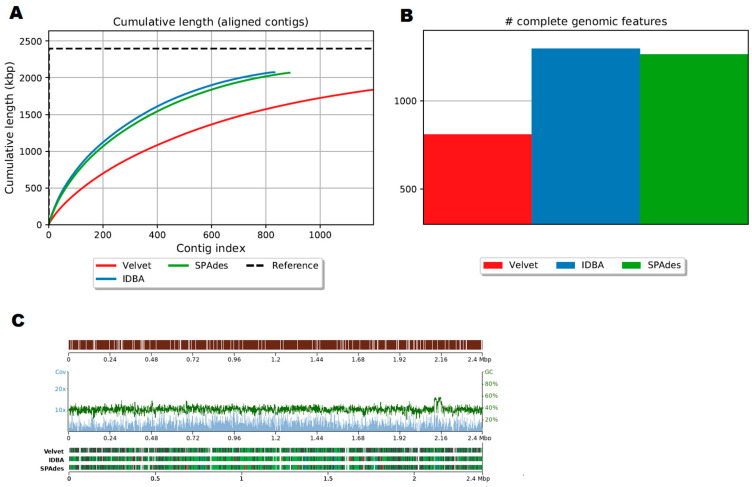
Efficiency of the de novo assembly of the *Ornithobacterium rhinotracheale* (ORT) genome obtained from the clinical sample from short-read datasets using the Velvet, SPAdes, and IDBA-UD genome assemblers. (**A**) Comparison of cumulative length of aligned contigs generated by Velvet, Spades, and IDBA-UD assemblers for clinical samples mapped to the ORT-UMN 88 (GenBank acc. no CP006828) reference genome sequence. (**B**) Comparison of complete genomic futures obtained by Velvet, SPAdes, and IDBA-UD de novo assemblers mapped along the ORT-UMN 88 reference genome. (**C**) The contig alignment viewer for Velvet, IDBA-UD, and SPAdes assemblies in the dataset. Contigs aligned to the ORT-UMN 88 (GenBank acc. no CP006828) reference genome sequence. From top to bottom: assembly to reference genome with gaps, detailed read coverage and GC content, and assembly overview. Correct contigs are colored in green and blue, whereas misassemblies are colored in red and orange.

**Figure 4 vetsci-12-00162-f004:**
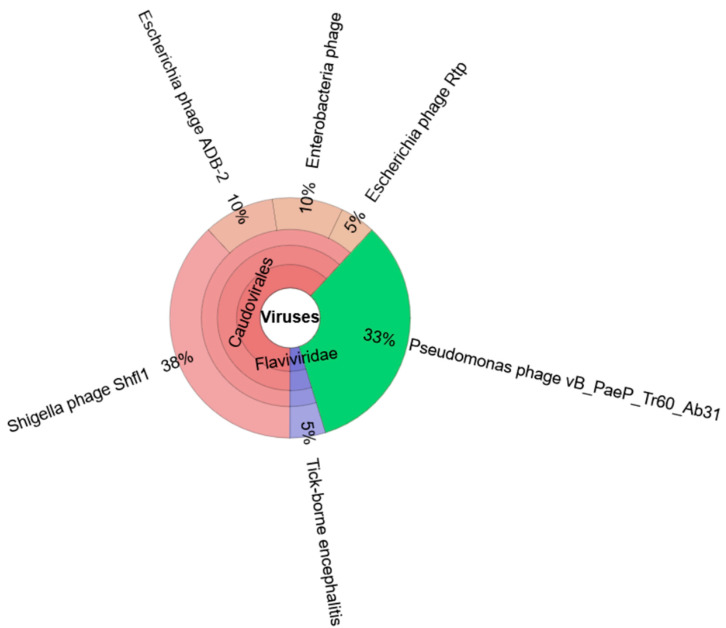
Taxonomic distribution of 72 de novo-assembled contigs (237,435 bp in total), selected based on size (minimum contig length of 1500 bp) and coverage (minimum contig coverage of 1000 bp) classified by the Kraken metagenomics tool using the viral database installed on a Galaxy platform (Galaxy Version 1.3.0). A total of 21 contigs were assigned to taxonomic labels based on k-mers species identification. Out of the total contigs used for analysis, eight were assigned as Shigella phage ShFl1 type and seven contigs were assigned as *Pseudomonas* phage type.

**Figure 5 vetsci-12-00162-f005:**
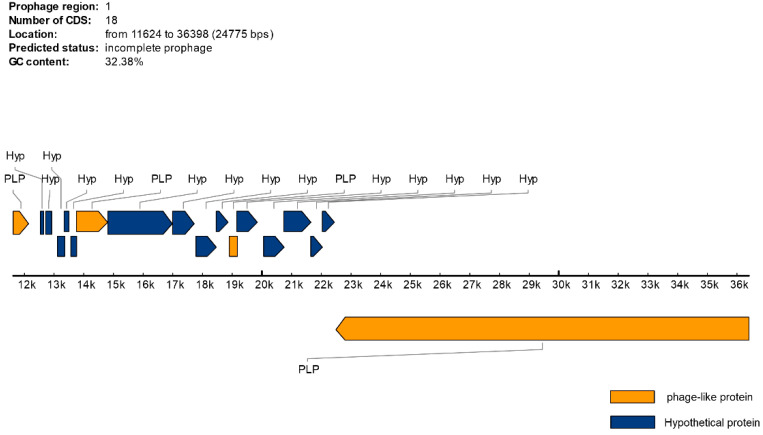
Phage-related proteins identified by PHAST. The diagram was annotated by PHAST-prophage database (Last update: 8 February 2017). The PLP annotated by PHAST represents putative HNH homing endonuclease, gp42, Bbp48, and virion RNA polymerase. In addition, nine hypothetical proteins in the region were found. An incomplete prophage of 24.7 kb was annotated. The proteins above the black line indicate genome positions that are encoded 5′ to 3′, while the proteins under it are encoded 3′ to 5′. A prophage sequence was submitted to GenBank under accession number MT025940.

**Table 1 vetsci-12-00162-t001:** Detailed statistics of the de novo assembly of clinical chicken samples in a single run that allowed for the detection and assembly of the *Ornithobacterium rhinotracheale* (ORT) genome. De novo assembly was performed using the Velvet de novo Assembler installed in the Illumina BaseSpace app (Version 1.0.0), SPAdes (version 3.10.1), and IDBA-UD (version 1.1.3); quality was assessed using QUAST (version 5.0.2).

Statistics	Velvet	IDBA-UD	Spades
#contigs (≥0 bp)	4960	3516	7895
#contigs (≥1000 bp)	806	815	825
#contigs (≥5000 bp)	26	100	94
#contigs (≥10,000 bp)	3	15	13
#contigs (≥25,000 bp)	1	1	1
#contigs (≥50,000 bp)	0	1	1
Largest contig (bp)	33,840	68,857	68,857
Total length (bp)	2,255,318	3,084,290	3,551,185
Total length (≥0 bp)	3,212,466	3,700,796	5,581,298
Total length (≥1000 bp)	1,694,308	2,310,795	2,271,117
Total length (≥5000 bp)	204,625	823,673	754,287
Total length (≥10,000 bp)	66,950	243,893	211,756
Total length (≥25,000 bp)	33,840	68,857	68,857
Total length (≥50,000 bp)	0	68,857	68,857
N50	1724	2438	1801
N75	1002	992	740
L50	378	314	450
L75	804	818	1291
GC (%)	40.07	42.02	43.06

# total number of contigs in the assembly; N50 is the length for which the collection of all contigs of that length or longer covers at least half an assembly; L50 is the number of contigs equal to or longer than N50.

**Table 2 vetsci-12-00162-t002:** KmerFinder 3.1 top three results for chicken clinical samples. De novo-assembled by Velvet, IDBA-UD, or SPAdes contigs were used as an input for species identification.

Assembler	Species	Score	Expected	Query Coverage (%)	Template Coverage (%)	Depth	Total Query Coverage (%)	Total Template Coverage (%)	*p*-Value	q-Value
Velvet	*Ornithobacterium rhinotracheale*	66,576	3	64.91	86.18	0.84	64.91	86.18	1.00 × 10^−26^	66,565.24
	*Streptococcus pluranimalium*	2482	17	2.42	3.28	0.03	2.44	3.3	1.00 × 10^−26^	2428.65
	*Bacteroides fragilis*	1807	42	1.76	1.02	0.01	1.83	1.05	1.00 × 10^−26^	1682.62
IDBA-UD	*Ornithobacterium rhinotracheale*	70,264	5	59.14	91.46	0.88	59.14	91.46	1.00 × 10^−26^	70,248.92
	*Streptococcus pluranimalium*	4270	19	3.59	5.64	0.05	3.61	5.66	1.00 × 10^−26^	4210.97
	*Corynebacterium pseudopelargi*	2815	19	2.37	3.87	0.04	2.4	3.89	1.00 × 10^−26^	2757.52
SPAdes	*Ornithobacterium rhinotracheale*	70,101	12	39.78	91.74	0.88	39.78	91.74	1.00 × 10^−26^	70,064.06
	*Streptococcus pluranimalium*	9312	25	5.28	12.4	0.12	5.3	12.42	1.00 × 10^−26^	9234.9
	*Bacteroides fragilis*	9161	66	5.2	4.73	0.05	5.25	4.77	1.00 × 10^−26^	8962.4

**Table 3 vetsci-12-00162-t003:** Multi-locus sequence typing analysis of *Ornithobacterium rhinotracheale* (ORT) genome recovered from a representative chicken oropharyngeal swab sample.

GENE	%IDENTITY	Alignment Length ^1^	Database Allele Length	GAPS	BEST MATCH
*adk*	100	393	393	0	adk_1
*aroe*	100	489	489	0	aroE_1
*fumc*	100	489	489	0	fumC_1
*gdha*	100	480	480	0	gdhA_1
*mdh*	100	519	519	0	mdh_1
*pgi*	100	492	492	0	pgi_1
*pmi*	100	489	489	0	pmi_1

^1^ The total number of nucleotides considered when comparing gene sequences at a specific gene locus.

**Table 4 vetsci-12-00162-t004:** Resistance gene analysis of the ORT-positive sample. Analysis was performed using BLAST-based approaches using The Comprehensive Antibiotic Resistance Database (CARD 3.0.2) on web portal—Resistant Gene Identifier (RGI 5.0.0). Open reading frame (ORF) prediction was performed using Prodigal, homolog detection was performed using DIAMOND, and Strict significance was performed based on CARD-curated bitscore cut-offs.

Gene	Detection Criteria	AMR Gene Family	Drug Class	Resistance Mechanism	% Identity of Matching Region	%Length of Reference Sequence	Bitscore	Cut-Off Value
tetQ	protein homolog model	tetracycline-resistant ribosomal protection protein	tetracycline antibiotic	antibiotic target protection	97.02	97.11	1234.6	1200
tet(Z)	protein homolog model	major facilitator superfamily (MFS) antibiotic efflux pump	tetracycline antibiotic	antibiotic efflux	99.25	34.64	240.7	700
tet(Z)	protein homolog model	major facilitator superfamily (MFS) antibiotic efflux pump	tetracycline antibiotic	antibiotic efflux	98.7	20.05	152.5	700
tetO	protein homolog model	tetracycline-resistant ribosomal protection protein	tetracycline antibiotic	antibiotic target protection	98.31	28.01	346.3	1200
AAC(3)-IV	protein homolog model	AAC(3)	aminoglycoside antibiotic	antibiotic inactivation	99.55	89.15	458.8	400

**Table 5 vetsci-12-00162-t005:** Detailed statistics for the BWA-MEM reference mapping of sequencing reads to *Ornithobacterium rhinotracheale* ORT-UMN 88 (GenBank CP006828). The quality was assessed using QUAST (version 5.0.2).

Statistics	Velvet	IDBA-UD	SPAdes
Genome			
Genome fraction (%)	76.251	86.401	86.117
Number of genomic features	809 complete 1086 partial	1296 complete 743 partial	1264 complete 767 partial
Largest alignment	8490	21,068	14,500
Total aligned length	1,837,908	2,075,141	2,068,563
NG50 ^1^	1605	3340	3125
NA50 ^2^	1553	2098	1300
NGA50 ^2^	1466	2894	2721
LG50 ^3^	421	193	207
LA50 ^2^	428	365	548
LGA50 ^2^	475	226	246
Reads mapping			
Number of mapped	64,195	72,130	75,136
Mapped (%)	53.47	60.32	62.88
Number of properly paired	57,814	67,784	70,416
Properly paired (%)	48.16	56.68	58.93
Singletons (%)	1.59	1.27	1.84
Misjoint mates (%)	3.07	2.06	1.88
Avgerage coverage depth	6	5	4
Coverage ≥ 1× (%)	98.77	98.47	97.85
Coverage ≥ 5× (%)	53.04	41	35.41
Coverage ≥ 10× (%)	12.08	9.11	7.99
Number of misassemblies	1	12	15
Misassembled contigs length	1570	53,662	83,821
Number of fully unaligned contigs	422	1106	1918
Fully unaligned length	415,075	996,853	1,476,749
Mismatches			
Number of mismatches	557	807	993
Number of indels	25	41	82
Indels length	185	165	508
Number of N’s	127	0	0
Number of N’s per 100 kbp	5.63	0	0

^1^ N50 is the length for which the collection of all contigs of that length or longer covers at least half an assembly; L50 is the number of contigs equal to or longer than N50; ^1^ NG50 is the length of the scaffold that covers at least 50% of the genome; L50 is the number of contigs equal to or longer than N50; ^3^ LG50 refers to the minimum length, in the set of longest contigs whose length sums to half the genome length (G for genome); ^2^ NA50, ^2^ NGA50, ^2^ LGA50 (“A” stands for “aligned”) are similar to the corresponding metrics without “A”, but in this case aligned blocks instead of contigs are considered. Aligned blocks are obtained by breaking contigs at misassembly events and removing all unaligned bases.

**Table 6 vetsci-12-00162-t006:** List of genes predicted using Prophage Hunter. The longest de novo-assembled contig of 68 kbp was used as an input for prophage screening.

Gene ID	Protein Length (aa)	NCBI	Pfam	InterPro
gene 34	957	putative endolysin	Transglycosylase SLT domain	Transglycosylase SLT domain 1
gene 35	187	Hypothetical protein BADFISH 36	N/A	N/A
gene 36	207	hypothetical protein SBVP1 0076	N/A	N/A
gene 37	1125	hypothetical protein CPT_Pollock62	N/A	N/A
gene 38	559	gp083	N/A	N/A
gene 39	214	hypothetical protein CPT Pollock64	N/A	N/A
gene 40	420	N4 gp56-like protein	N/A	N/A
gene 41	426	gp086	N/A	N/A
gene 42	113	hypothetical protein DSS3P2_gp68	N/A	N/A
gene 43	811	portal protein	N/A	N/A
gene 44	150	putative endolysin	D-alanyl-D-alanine carboxypeptidase	Peptidase M15C
gene 45	111	membrane protein	N/A	N/A
gene 46	138	hypothetical protein	N/A	N/A
gene 47	133	ORF43	N/A	N/A
gene 48	83	gp29	N/A	N/A
gene 49	67	virion RNA polymerase	N/A	N/A
gene 50	166	hypothetical protein PaBG 00240	N/A	N/A
gene 51	174	HtpB	N/A	N/A
gene 52	187	putative tail measure protein	N/A	N/A
gene 53	719	baseplate wedge	N/A	N/A
gene 54	539	hypothetical protein SUAG 00002	N/A	N/A
gene 55	181	hypothetical protein Av05 0058	N/A	N/A

N/A: not applicable.

**Table 7 vetsci-12-00162-t007:** List of genes predicted using PHAST. The longest de novo-assembled contig of 68 kb was used as an input for prophage screening.

CDS	BLAST HIT	E-VALUE
1	PHAGE Entero GEC 3S NC 025425: Putative HNH homing endonuclease; PP 00015; (gi754380660)	4.00 × 10^−9^
2	hypothetical; PP 00016	N/A
3	hypothetical; PP 00017	N/A
4	hypothetical; PP 00018	N/A
5	hypothetical; PP 00019	N/A
6	hypothetical; PP 00020	N/A
7	PHAGE Escher N4 NC 008720: gp42; PP 00021; phage (gi119952219)	2.00 × 10^−60^
8	PHAGE Entero EcP1 NC 019485: hypothetical protein; PP 00022; phage (gi418489348)	0
9	PHAGE Pseudo pYD6 A NC 020849: hypothetical protein; PP 00023; phage (gi472340759)	4.00 × 10^−61^
10	PHAGE Pseudo KPP21 NC 029017: hypothetical protein; PP 00024; phage (gi971766098)	2.00 × 10^−25^
11	PHAGE Entero EcP1 NC 019485: hypothetical protein; PP 00025; phage (gi418489351)	2.00 × 10^−24^
12	PHAGE Bordet BPP 1 NC 005357: Bbp48; PP 00026; phage (gi41179408)	9.00 × 10^−25^
13	PHAGE Erwini vB EamP Frozen NC 031062: hypothetical protein; PP 00027; phage (gi100065)	2.00 × 10^−5^
14	hypothetical; PP 00028	N/A
15	hypothetical; PP 00029	N/A
16	hypothetical; PP 00030	N/A
17	hypothetical; PP 00031	N/A
18	PHAGE Entero EcP1 NC 019485: virion RNA polymerase; PP 00032; phage (gi418489353)	4.00 × 10^−69^

N/A: not applicable.

**Table 8 vetsci-12-00162-t008:** A list of the closest related phages predicted by Prophage hunter.

Candidate ID	Subject ID	Subject Name	Identity	Query Coverage	E Value	Bitscore
Prophage	KF669658.1	Acinetobacter phage Presley	65%	4%	8.73 × 10^−39^	168
	KC139520.1	Salmonella phage FSL SP-076	66%	4%	8.73 × 10^−39^	168
	KC139517.1	Salmonella phage FSL SP-058	66%	8%	5.51 × 10^−35^	156
	KM236242.1	Escherichia phage Pollock	69%	3%	3.48 × 10^−31^	143
	KY065148.1	Vibrio phage JSF3	72%	2%	4.24 × 10^−30^	140

## Data Availability

The prophage sequence was submitted to GenBank under accession number MT025940 and BioProject: PRJNA55733. All read sequences were deposited in the Sequence Read Archive (SRA) of the NCBI (http://www.ncbi.nlm.nih.gov/sra) on February 2022 under accession number SRR9903547. The prophage sequences were submitted under GenBank accession number MT025940.

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
