# Peer review of "Whole-Genome Shotgun Sequencing from Chicken Clinical Tracheal Samples for Bacterial and Novel Bacteriophage Identification"

_vetsci, 2025, doi:10.3390/vetsci12020162_

Round 1
Reviewer 1 Report
Comments and Suggestions for Authors
The study employed whole-genome shotgun sequencing (sWGS) technology to directly identify pathogens in chicken clinical samples, eliminating the need for traditional bacterial culture. This approach significantly reduced the diagnostic time from several weeks to just a few days, providing robust technical support for rapid diagnosis and treatment. It also introduced new research ideas and methods to the field of clinical microbiology, demonstrating innovation. However, the study could be improved in terms of sample size, clinical relevance, and phage function research to enhance its depth and breadth, better serving clinical practice and scientific research.
Regarding line 221: "Velvet assembled 1.8 Mb of contigs, whereas IDBA-UD, and SPAdes assembled 2.0 kb of contigs." The unit "2.0 kb" should be "2.0 Mb" to maintain consistency with the units used earlier in the text. The corrected sentence should read: "Velvet assembled 1.8 Mb of contigs, whereas IDBA-UD and SPAdes assembled 2.0 Mb of contigs."
Author Response
Comments and Suggestions for Authors
The study employed whole-genome shotgun sequencing (sWGS) technology to directly identify pathogens in chicken clinical samples, eliminating the need for traditional bacterial culture. This approach significantly reduced the diagnostic time from several weeks to just a few days, providing robust technical support for rapid diagnosis and treatment. It also introduced new research ideas and methods to the field of clinical microbiology, demonstrating innovation. However, the study could be improved in terms of sample size, clinical relevance, and phage function research to enhance its depth and breadth, better serving clinical practice and scientific research.
Regarding line 221: "Velvet assembled 1.8 Mb of contigs, whereas IDBA-UD, and SPAdes assembled 2.0 kb of contigs." The unit "2.0 kb" should be "2.0 Mb" to maintain consistency with the units used earlier in the text. The corrected sentence should read: "Velvet assembled 1.8 Mb of contigs, whereas IDBA-UD and SPAdes assembled 2.0 Mb of contigs."
Author`s reply: thank you for your suggestion and comments about our manuscript. This was corrected.
Reviewer 2 Report
Comments and Suggestions for Authors
General comment
The work has been done well and it would be great to publish it in this journal. However, its publication is only possible once the issue mentioned below have been addressed.
- Title
L2-4: The title is misleading... How do you talk of “Whole genome short gun sequencing from unknown samples …" when you actually know them and even clearly describe them briefly at L23-4("… chicken clinical tracheal ...") and L92 ("… Tracheal swabs ..."). Please rectify this error.
- Introduction
L43-53: I don't this part is really important for this work. Would you mind getting rid of it?
L50: GenBank now accept of… Please delete " … of ..."
- Materials & Methods
L92: “… BH broth…” Provide the source of this medium (manufacturer's name, city, and country of origin/ catalogue number of the product).
- Results
L381: "... Podoviridae..." this family name currently doesn't exist and ceased being used about three years back (2022). Please visit the ICTV website for further details.
L383: “... to other phage genes such as Escherichia…" Please italicize “.. Escherichia …
Author Response
We would like to thank you for your insights to improve the manuscript.
General comment
The work has been done well and it would be great to publish it in this journal. However, its publication is only possible once the issue mentioned below have been addressed.
- Title
L2-4: The title is misleading... How do you talk of “Whole genome short gun sequencing from unknown samples …" when you actually know them and even clearly describe them briefly at L23-4("… chicken clinical tracheal ...") and L92 ("… Tracheal swabs ..."). Please rectify this error.
Author`s reply: The title was changed to reflect the work done here.
- Introduction
L43-53: I don't this part is really important for this work. Would you mind getting rid of it?
L50: GenBank now accept of… Please delete " … of ..."
Author`s reply: thank you, it is a valid point. It does not add much to the aim of this study.
- Materials & Methods
L92: “… BH broth…” Provide the source of this medium (manufacturer's name, city, and country of origin/ catalogue number of the product).
Author`s reply: missing information add to the main test
- Results
L381: "... Podoviridae..." this family name currently doesn't exist and ceased being used about three years back (2022). Please visit the ICTV website for further details.
Author`s reply: this was corrected.
L383: “... to other phage genes such as Escherichia…" Please italicize “.. Escherichia …
Author`s reply: that was changed.
Reviewer 3 Report
Comments and Suggestions for Authors
The authors collected chicken tracheal swab samples for the detection of potential pathogens. Whole-genome shotgun sequencing was used as the major experimental approach, along with other computational methods. The results confirmed the infection of Ornithobacterium rhinotracheale (ORT) and a N4-like prophage. Overall I believe this work is solid and it may benefit the ones that would like to use sequencing for pathogen detection in clinical samples. I only have a few minor questions/suggestions:
Is it possible to specifically tell what the PLPs are in Figure 5? It would be much appreciated if the authors could annotate these genes since they are discussed in detail in page 16.
How did the authors determine the prophage identified in the sample should be classified as a N4-like prophage (line 293-294)?
There seems to be no known phages that infect ORT. The co-existence of ORT and the prophage identified might indicate that this phage infects ORT. This possibility could be discussed in the Discussion.
Author Response
We would like to thank you for your insights to improve the manuscript
The authors collected chicken tracheal swab samples for the detection of potential pathogens. Whole-genome shotgun sequencing was used as the major experimental approach, along with other computational methods. The results confirmed the infection of Ornithobacterium rhinotracheale (ORT) and a N4-like prophage. Overall I believe this work is solid and it may benefit the ones that would like to use sequencing for pathogen detection in clinical samples. I only have a few minor questions/suggestions:
Is it possible to specifically tell what the PLPs are in Figure 5? It would be much appreciated if the authors could annotate these genes since they are discussed in detail in page 16.
Author`s reply: The proteins showed on the Figure 5 are explained in the table 7. We agree, however, that this might not be clear. We add missing information to the figure caption.
How did the authors determine the prophage identified in the sample should be classified as a N4-like prophage (line 293-294)?
Author`s reply: This sentence was correct.
“The prophage sequence obtained from a chicken tracheal swab sample discovered in this was characterized as most closely related to an N4-like Prophage Chicken Bacterial Metagenome based on BLAST analysis. All ORFs predicted were next manually reanalyzed by a BLAST search. BLAST analyses of annotated ORFs from the putative prophage sequence resulted in a variety of potential bacterial encoded and principally bacteriophage encoding genes indicating that the sequenced DNA was a potential prophage”
There seems to be no known phages that infect ORT. The co-existence of ORT and the prophage identified might indicate that this phage infects ORT. This possibility could be discussed in the Discussion.
Author`s reply: Thank you for suggestion, we add information and citation to the discussion part (392-396).